# Comparative Analysis of the Chloroplast Genome of *Sicyos angulatus* with Other Seven Species of Cucurbitaceae Family

**DOI:** 10.3390/genes14091776

**Published:** 2023-09-08

**Authors:** Muniba Kousar, Joonho Park

**Affiliations:** Department of Fine Chemistry, Seoul National University of Science and Technology, 232-Gongneung-ro, Nowon-gu, Seoul 01811, Republic of Korea

**Keywords:** *Sicyos angulatus*, comparative chloroplast genome analysis, Cucurbitaceae family, phylogenetic relationship

## Abstract

*Sicyos angulatus* (SA) is an annual plant from the Cucurbitaceae family that is native to the eastern part of North America. This study aims to assemble and annotate the chloroplast genome of *S. angulatus*, and then compare it with plastomes of the other species representing the Cucurbitaceae family. The chloroplast genome size of *S. angulatus* is 154,986 bp, including a pair of inverted repeats (IR) of 26,276 bp, and small single-copy region (SSC) of 18,079 bp and large single-copy region (LSC) of 84,355 bp. Compared to other Cucurbitaceae species, the chloroplast genome of *S. angulatus* is almost 4222 bp smaller than the plastome *Gynostemma pentaphyllum*. All other seven species have an identical set of tRNA (37), except *Citrullus laevigata*, which contains 36 tRNA. The IRa/LSC junction in all eight species is located upstream of *rpl2* and downstream of *trnH* gene. Moreover, variation in the size of the gene and the presence of pseudogene *ycf1* has been seen because of the IR contraction and expansion. The highest number of tandem repeats was seen in *G. pentaphyllum*, and then *Corynocarpus leavigata*. The sequence divergence analysis and topology of the phylogenetic tree indicate that *S. angulatus* is more similar to genus *Citrullus* as compared to genus *Gynostemma*. These findings contribute to developing the genomic marker for the purpose of future genetic studies.

## 1. Introduction

*Sicyos Angulatus* (SA), commonly called Bur Cucumber, is an annual vine plant in the gourd family, Cucurbitaceae, that is native to the eastern part of North America. The Cucurbitaceae family consists of approximately 965 species and around 95 genera [1]. *S. angulatus* is also known as invasive plant and was introduced to South Korea as a parent stock for cucumber cultivation in the 1980s [2,3]. Indeed, reports of the *Sicyos angulatus* have been found in India and other countries in Asia [4]. It is adapted to wet habitats, river floodplains, and colonizes opened habitats, like fencerows, roadsides, and woodland borders. It is found in various eastern parts of the Rocky Mountains, Canada, Mexico, and eastern Asia [5,6]. The genus *Sicyos* contains almost 40 species, and it belongs to the most diversified genera in the Cucurbitaceae family [7]. The features of chloroplast genomes (cp) of some Cucurbitaceae species have been investigated [8]. Moreover, one study revealed the potential therapeutic effect of *Sicyos Angulatus* in acute liver disorder and an atherosclerosis mouse model [9,10]. SA extracts also show an inhibitory effect of flavone glycosides on hepatic lipid accumulation [11]. In the latest study, the anti-obesity effect of *S. angulatus* in HFD-induced (High-Fat diet) obese mice by inhibiting the accumulation of fat has been reported [12]. These findings emphasize *Sicyos angulatus*’s multifaceted nature, including its ecological adaptation, agricultural significance, and medicinal potential. Future studies of this plant species and its associated molecular features, such as the chloroplast genome, can reveal important details about its evolutionary background, genetic diversity, and potential uses in a variety of fields, including plant breeding programs and research on medications.

The chloroplast (cp) arose from an endosymbiotic relationship between photosynthetic bacteria and a non-photosynthetic host [13]. Moreover, molecular studies show that it still contains the remnants of the eubacterial genomes, but during evolution, most of their genes transferred to the nucleus [14,15]. The chloroplast genome has been used widely for multiples purposes, such as source of molecular markers, barcode identification, phylogenetic analysis, and providing a wealth of other valuable feature that make it an interesting and highly desirable object of study in molecular research. The chloroplast genome has a remarkable variety of characteristics that contribute to its importance, including its highly conserved nature, compact size, ease of extraction, relatively low mutation rates, abundant presence within plant cells, and its capacity to encode crucial genes involved in photosynthesis and other crucial cellular process. These numerous characteristics further increase the value and appeal of the chloroplast genome as a flexible tool in different scientific studies [16,17,18]. The chloroplast is a photosynthetic organelle that plays a vital role in the synthesis of starch, amino acids, and fatty acids [19,20]. In general, the chloroplast genome size usually ranges from 120 to 160 kb; it exhibits a highly conserved circular quadripartite structure, which includes four major regions: a large single copy (LSC), small single copy (SSC), and two inverted repeat regions (IR) [21,22]. Moreover, the structural characteristics of plastome are highly common among land plants. Additionally, their usage as significant models for evolutionary studies is made possible by uniparental inheritance, low rates of substitution, and their small size as compared to that of the nuclear genome.

The number of sequenced chloroplast genomes has soared in correlation with the advancements made in second-generation high-throughput sequencing technology [23]. Based on chloroplast genomic information, phylogenetic relationships for many plant species as well as the reclassification of different plant taxa have been inferred [24]. Therefore, the exploration of the *Sicyos* cp genome may provide a new breakthrough to solve the issues of the genus’ evolution and its phylogenetic relationship with other taxa. By comparing cp genomes of different *Sicyos* species and related taxa, researchers can find genetic variants, sequence similarities, and structural rearrangement that can reveal important information about their evolutionary history.

In this research, the complete chloroplast genome of *Sicyos angulatus* and the other seven species of Cucurbitaceae family was chosen as a research target. The aims of this study are (I) to characterize the structure of the newly sequenced chloroplast genome of *S. angulatus*, (II) to perform the comparative analysis of the cp genomes of *S. angulatus* and selected representatives of Cucurbitaceae family, and (III) to reconstruct phylogenetic relationships within the Cucurbitaceae family based on plastome sequences for selected species. These data will be helpful to develop a genomic marker for the purpose of future genetic studies.

## 2. Materials and Methods

### Gene Annotation, Sequence Alignment, and Repeat Prediction

The complete chloroplast sequences of eight species were obtained from GenBank for comparative analysis with the following accession numbers: *S. angulatus* (NC_062884.1), *Corynocarpus laevigata* (NC_014807.1), *Gynostemma pentaphyllum* (NC_029484.1), *Gynostemma pentagynum* (NC_036136.1), *Gynostemma longipes* (NC_036140.1), *Citrullus naudinianus* (NC_058581.1), and *Sicyos edulis* (MT and *Citrullus ecirrhosus* (NC_058582.1).

Geneious prime software (Geneious Prime v.2023.2.1) was used to annotate the chloroplast genome, and the manual evaluation of annotation results was conducted. OGDRAW software version number 1.3.1 was used to plot the circular DNA map [25]. Pairwise sequence alignment was performed with online comparison tool mVISTA [26] using the annotation of *S. angulatus* as reference with the Shuffle-LAGAN mode and a 70% cut-off identity [27]. Tandem repeats were analyzed by using Tandem Repeat Finder version 4.09 with the following parameters, 2 for alignment parameters match, 7 for mismatch, and 7 for indels [28]. The comparison of the LSC/IRB/SSC/IRA boundaries among eight species was performed by using an online tool IRSCOPE (https://irscope.shinyapps.io/irapp/ accessed on 1 August 2023) [29]. Microsatellites (SSRs) analysis was performed using Krait software (Krait V1.3.3, Microsatellite identification and Primer Design; https://krait.biosv.com/en/latest/ accessed on 3 August 2023) with the following parameters: 10 for mono-nucleotide, 5 for di-nucleotide, 4 for tri-nucleotide, 3 for tetra-nucleotide, 3 for penta-nucleotide, and 3 for hexa-nucleotide repeats, and the motif standardization level was set to level 3 [30]. To investigate the phylogenetic relationship, CDs sequence of 25 species were extracted from whole cp genome and aligned with MAFFT v.7.450 [31]. The phylogenetic tree was constructed using the Neighbor-Joining method and the Tamura–Nei genetic distance model with geneious software; *Begonia versicolor* was selected as an outgroup. For the clade support, a 100 bootstrap value was used.

## 3. Results

### 3.1. General Characteristics of the Sicyos angulatus Chloroplast Genome

*S. angulatus* has a complete CP genome with a length of 154,986 bp and displays a quadripartite circular structure. It consists of a pair of IR regions, each 26,276 bp in length, separated by an LSC region and an SSC region with lengths of 84,355 bp and 18,079 bp, respectively. The complete genome contains a total of 129 genes, including 84 protein-coding gene (PCGs), 37 tRNA, and 8 rRNA genes. The LSC region contains 83 genes, while the SSC region consists of 14 genes. Additionally, 17 genes are duplicated in the IRs (Figure 1). The protein-coding gene accounts for 51.24% (79,419 bp) of the total genome, whereas the remaining regions are composed of rRNA, tRNA, and intron and intergenic spaces.

In addition, we observed 22 intron-containing genes, out of which 19 genes contained one intron (11 protein-coding genes and 8 tRNA genes) and 2 genes contained two introns (*pafl* and *clpP1*). The total GC content of cpDNA was 37.2%, but we observed distinct differences between the three regions of cp genome when this feature was considered. The highest GC content exhibited in the IR regions (42.8%) were followed by the LSC and SSC regions (35.1% and 31.0%, respectively).

### 3.2. Comparative Analysis with Other Chloroplast Genomes from the Cucurbitaceae Family

The *S. angulatus* chloroplast genome was compared to seven other species of the Cucurbitaceae family.

#### 3.2.1. Genome Size and Gene Content

The complete chloroplast genome of *S. angulatus* is compared to other seven species of the Cucurbitaceae family.

The complete Cucurbitaceae CP genome has highly conserved structures, regarding its gene content. Each chloroplast genome encodes almost 133 genes, except *C. laevigata* and *S. angulatus* which possess 128 and 129 genes, respectively. However, the chloroplast genome of *S. edulis* encodes a low total gene content, approximately 123 genes. Moreover, they have 37 tRNA (except *C. laevigata*), 8 rRNA, and two genes with double introns. A total of 19 genes in *G. longipes*, *G. pentaphyllum*, *C. naudinianus*, and *S. angulatus*, 20 genes in *C. leavigata*, *C. ecirrhosus*, and 21 genes in *G. pentagynum* contain single introns, However, two genes contain double introns (*clpP* and *ycf3).*

The CP genome size of *G. pentaphyllum* is the largest among the eight studied genomes (159,208 bp), and it is 4222 bp longer than the plastomes of *S. angulatus* (Table 1).

Only *C. leavigata* lacks *trnfM-CAU* gene, although it has one *trnl-CAU*, andone additional *trnM-CAU* gene. All rRNAs (*rrn4.5*, *rrn5*, *rrn16*, and *rrn23*) are in the IR region. In addition to this, in *C. leavigata* and *G. pentaphyllum*, some of duplicated genes have not same sequence, consisting of a total of 21 and 15 duplicated genes, respectively.

All the species share seventy-four protein coding genes, except for five protein coding genes, as shown in Table 2. Genes without any mark are single-copy genes. The *ndhA* is a single-copy gene, but it is duplicated in *C. ecirrhosus*. Moreover, there are differences in the annotation as well, like in *S. angluatus*, *ycf3* and *ycf4* are annotated as *pafl* and *pafll*, respectively. 

#### 3.2.2. Sequence Divergence, Tandem Repeats, and SSR Analysis

The many complete CP genome permitted us to evaluate the sequence variation among species. The divergence of the sequence in the CP genome among the eight species was plotted with a cut-off of 70% identity by using mVISTA. The genome of *S. angulatus* was set as the reference genome.

Figure 2 shows that most of the divergence was seen in both the LSC and SSC regions. The IR region and protein-coding genes are more conservative compared to the non-coding region, while some portions of coding region also contain fair observable divergences among *atpF*, *rpoC2*, *rpoC1*, *clpP*, *ndhA*, *petB*, and *psbM*. In comparison to genus *Gynostemma*, *S. angulatus* is more closely linked to *Sicyos edulis*, and *Citrullus* genus, according to the sequence divergence analysis.

Tandem repeats among the eight species are present along the intergenic spaces and coding sequence region. The repeat sequence in *G. pentaphyllum* and *C. laevigata* is the most frequent in the intergenic spaces 74 and 60, respectively, while the repeat sequence of *C. ecirrhosus* (13) is less frequent in the intergenic space (Table 3). However, among the repeats with a range of 25–45, *C. laevigata* has the highest number of repeats, and *S. edulis* shows second lowest number of repeats, followed by *C. ecirrhosus* (Figure 3A). The overall distribution of these genes throughout the intergenic space and coding sequence region is illustrated in Figure 3B in the form of a percentage.

SSR analysis revealed that among eight species, mononucleotide repeats were the most abundant, followed by the di-, tri-, tetra, penta-, and hexanucleotide repeats (Figure 4A). The highest percentage count of mononucleotide repeats was observed in *C. laevigata* (72.8%), followed by *S. edulis* (80.0%), and lowest percent count of mononucleotide repeats was in *G. pentagynum* (46.6%). In mononucleotide repeats, A was found to be abundant in *C. laevigata* (93 counts), followed by *S. edulis* (60 counts) and *S. angulatus* (49 counts) (Figure 4B). In di-nucleotide repeats, AT in tri-nucleotide AAT, in tetra-nucleotide AAAT, in penta-nucleotide AGGGG, and in hexanucleotide AAATGG is most abundant among eight species of the Cucurbitaceae family. Moreover, the SSRs length distribution for mono-nucleotide repeats is highest in *S. edulis* (78.2%), followed by *S. angulatus* (76.8%), *C. laevigata* (73.4%), *C. naudinianus* (71.2%), *C. ecirrhosus* (61.4%), *G. longipes* (46.5), *G. pentagynum* (42.9%), and *G. pentaphyllum* (39.7%) (Figure 4C).

#### 3.2.3. Contraction and Expansion of IRs

The comparison of the genes adjacent to the IR/SSC and IR/LSC boundaries of the analyzed plastomes shows a slight variation, as represented in Figure 5. In this study, the 46 bp and 19 bp of *rps19* genes span the LSC/IRb regions of *C. laevigata* and *G. pentagynum*, respectively, while the other species *rps19* has shares 2 bp with the LSC/IRb region, but in *S. edulis* species, *rps19* is completely located in the LSC region. Notable differences were observed on the junction of IRb/SSC (JSB). At this junction, *ycf1* was absent in *C. laevigata*, while the remaining six species contain *ycf1**ψ*** (pseudogene). Additionally, the *ndhF* gene was located at the IRb/SSC junction in *G. pentagynum* and *G. longipes*, *C. ecirhosus*, and *S. angulatus*, wherein 12 bp was located in the IRb region in *G. pentagynum* and *G. longipes*, while 7 and 6 bp were found in the IRb region in *C. ecirhosus* and *S. angulatus*, respectively. It is noteworthy to mention *ycf1*, which spans the SSC/IRa junction in all eight species. Moreover, on the JSB junction, the *ycf1* gene spans IRb/SSC border, except in *G. pentaphyllym*, wherein *ycf1* is entirely located in the IRb region. On the other hand, *ycf1* is absent in *S. edulis* species. *psbA* and *trnH* genes are located entirely in the LSC region. 

#### 3.2.4. Phylogenetic Analysis

To determine the phylogenetic relationship between *S. angulatus* and other species of Cucurbitaceae family, the Tamura–Nei method was used based on the plastome of all the species by using the CDs sequence from the complete cp genome sequence. All 25 species developed two branches with dedicated support using bootstrap values of 100% (Figure 6). The phylogenetic tree indicates that *Sicyos angulatus* is clustered with *Sicyos edulis*. In addition, *Sicyos* genus is close to the *Trichosanthes* genus. Also, the *Sicyos* genus is closely related to *Citrullus ecirrhosus* and *Citrullus naudinianus* (*Citrullus* genus). However, it can be seen that the *Sicyos genus* is more closer to *Gynostemma longipes*, *Gynostemma pentagynym*, *Gynostemma pentaphyllum* (*Gynostemm* genus), as well as *Corvynocarpus laevigata* from the Cucurbitaceae family.

## 4. Discussion

The chloroplast genome of *Sicyos angulatus* (SA) was assembled and annotated in this study, and these efforts shed light on the genetic makeup of this species and its relationship with other species in the Cucurbitaceae family [32]. We have obtained deeper knowledge of their genomic characteristics and evolutionary dynamics by contrasting the chloroplast genomes of SA with those of other species in the family. Moreover, this study offers a thorough grasp of the genomic size, gene content, and structural organization of the chloroplast genomes of the *Sicyos* and other genera belonging to the same family. 

The selected eight species were chosen to represent various genera within the Cucurbitaceae family and span a variety of taxonomic diversities. Because these species exhibit interesting ecological traits and phenotype diversity, they are useful for comparative analysis. It was observed that the *G. pentaphyllum* and *C. laevigata* chloroplast genome sizes were larger than that of *S. angulatus*. *C. ecirrhosus* has the largest coding size. However, *G. pentaphyllum* and *S. edulis* have the smallest coding size among the selected species. The notable causes of this fluctuation in the chloroplast genome size are intergenic region variance, the shrinkage and expansion of inverted repeats regions, and the loss of genes/introns, which are consistent with earlier reports [33]. This size discrepancy reflects possible changes in the structural and functional components of chloroplast genomes and points to genetic diversity within the family. For an understanding of the evolutionary trends and adaptive tactics within the Cucurbitaceae family, it is essential to comprehend these variances. 

The location of the boundaries between the four chloroplast sections is another crucial aspect of the chloroplast genome that is valuable for evolutionary investigation. Analyzing their contraction and expansion can provide an insight into how some taxa have evolved [34]. All eight species have similar placements of the IRa/LSC junction between the upstream and downstream regions of *rpls2* and *trnH* genes, suggesting that the family’s genomic organization is conserved. IR contraction or expansion affects the genes which are placed in a close vicinity to the LSC/IR/SSC borders. 

This study of eight complete chloroplast genomes (for *S. angulatus* and seven other representatives of the family Cucurbitaceae) gave us the chance to compare the sequence variance among the eight species. For visualizing the divergence in the CP genome sequence, we used the mVISTA tool. Similar to prior research, this study also found that noncoding areas were more diverged than the coding regions are [35,36]. The overall results demonstrated that *S. angulatus* is more like genus *Citrullus* than it is like the genus *Gynostemma*. This finding points to a more recent common ancestor or shared genetic inheritance between *S. angulatus* and *Citrullus* genera, indicating a closer evolutionary link between these two species.

Plant cp genomes frequently contain SSRs, which are frequently employed as molecular markers for polymorphism studies [37]. In this study, SSRs’ length distribution for the mono-nucleotide type showed significantly similar repeats in *S. angulatus* and *S. edulis*. However, the SSR count distribution and each standard motif in the mononucleotide were abundant in *C. laevigata* among eight species of the Cucurbitaceae family. Overall, a higher number of mononucleotides repeats as compared to those of di-hexa SSRs were observed in this study, which is consistent with one other study [38].

Previous studies have shown that the variation in size of the chloroplast genome can be due to the expansion and contraction of its region, mainly the IRs [39]. The nucleotide sequence was seen to be highly conserved in the IR regions, which can provide a powerful means to correct unavoidable mutations [40]. In addition, ycf1 lies on the IRa/SSC junction. The *ndhF* genes in *C. laevigata*, *G. pentaphyllum*, and *C. naudinianus* is completely on the SSC region, while 12 bp in *G. pentaphyllum* and *G. longipes*, 7 bp in *C. ecirrhosus*, and 6 bp in *S. angulatus* of *ndhF* overlap in the IRb region. No variation was seen at the JSA junction. *Ycf1* is located at the JSA junction. *trnH* is completely in the LSC region. In addition, *ycf1* is completely lacking in *sicyos edulis*. This finding is consistent with previous research that indicated *ycf1* is not onlylacking in grasses but also in cranberries [41].

Phylogenetic analysis demonstrated SA’s evolutionary relationship with the other examined species. Our phylogenetic tree indicated a very clear internal relationship between *Sicyos angulatus* and *Sicyos edulis*, which is similar to that in previous research [32]. Moreover, the tree suggests that the *Sicyos* genus is closely related to the *Citrullus* species: *Citrullus ecirrhosus* and *Citrullus naudinianus*. The phylogenetic differences observed in the tree are likely related to both physiological features and differences in the analyzed chloroplast genome. This study’s limitations in terms of its scope and emphasis may account for the lack of a thorough examination of the connections between the results of the phylogenetic analysis and other cp genome diversities. Instead of in-depth discussion about the link with other cp genome diversities, the primary goal may have been to investigate the cp genomes of chosen species and discover the phylogenetic relationships within the Cucurbitaceae family.

In conclusion, insights into the evolutionary relationships within the Cucurbitaceae family were gained through the assembly and annotation of the *Sicyos angulatus* chloroplast genome, as well as comparative research. Our knowledge of the genetic diversity and evolution of this plant family is influenced by the structural changes, gene makeup, and sequence divergence that have been found. The cp genome for *S. angulatus* may become resource to develop genomic markers, which can further help with future genetic research and support breeding initiatives, conservation efforts, and phylogenetic analyses of the family Cucurbitaceae.

## Figures and Tables

**Figure 1 genes-14-01776-f001:**
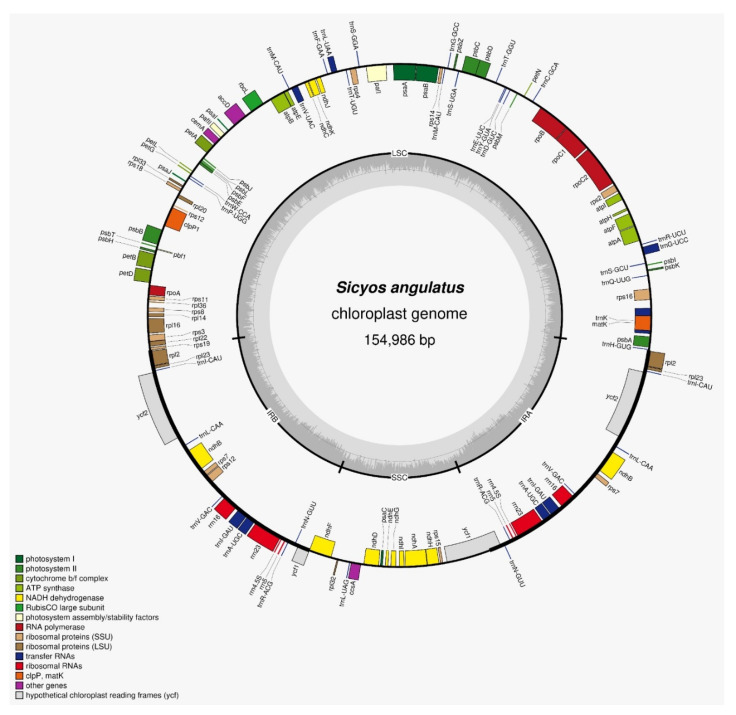
Map of *S. angulatus* chloroplast genome. The genes drawn inside are transcribed anticlockwise, while the outside genes are transcribed clockwise. The different colors represent genes from different functional groups.

**Figure 2 genes-14-01776-f002:**
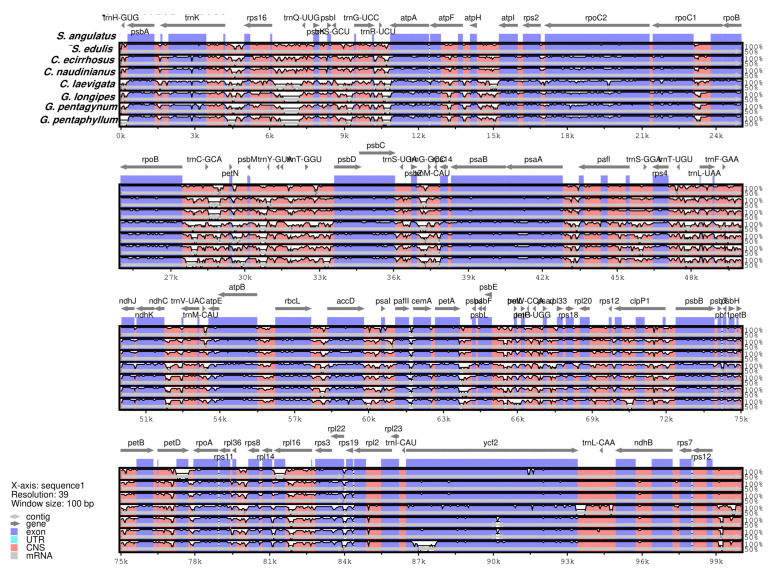
Sequence alignment plot for eight Cucurbitaceae species using mVISTA with *S. angulatus* genome as a reference. Gray arrows represent gene orientation and position. The *y*-axis point the percentage identity range of 50–100%. Red and blue bars indicate non-coding sequence (CNSs) and exons, respectively.

**Figure 3 genes-14-01776-f003:**
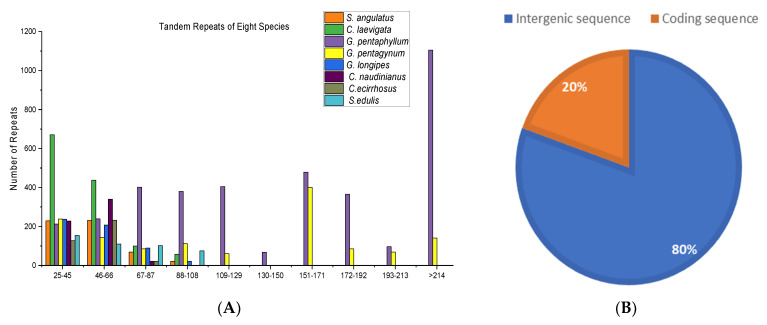
Length and distribution of tandem repeats using tandem repeat finder among eight species from Cucurbitaceae family. The number of repeats is shown on the *y*-axis (**A**). Location distribution of overall tandem repeats in percentage (**B**).

**Figure 4 genes-14-01776-f004:**
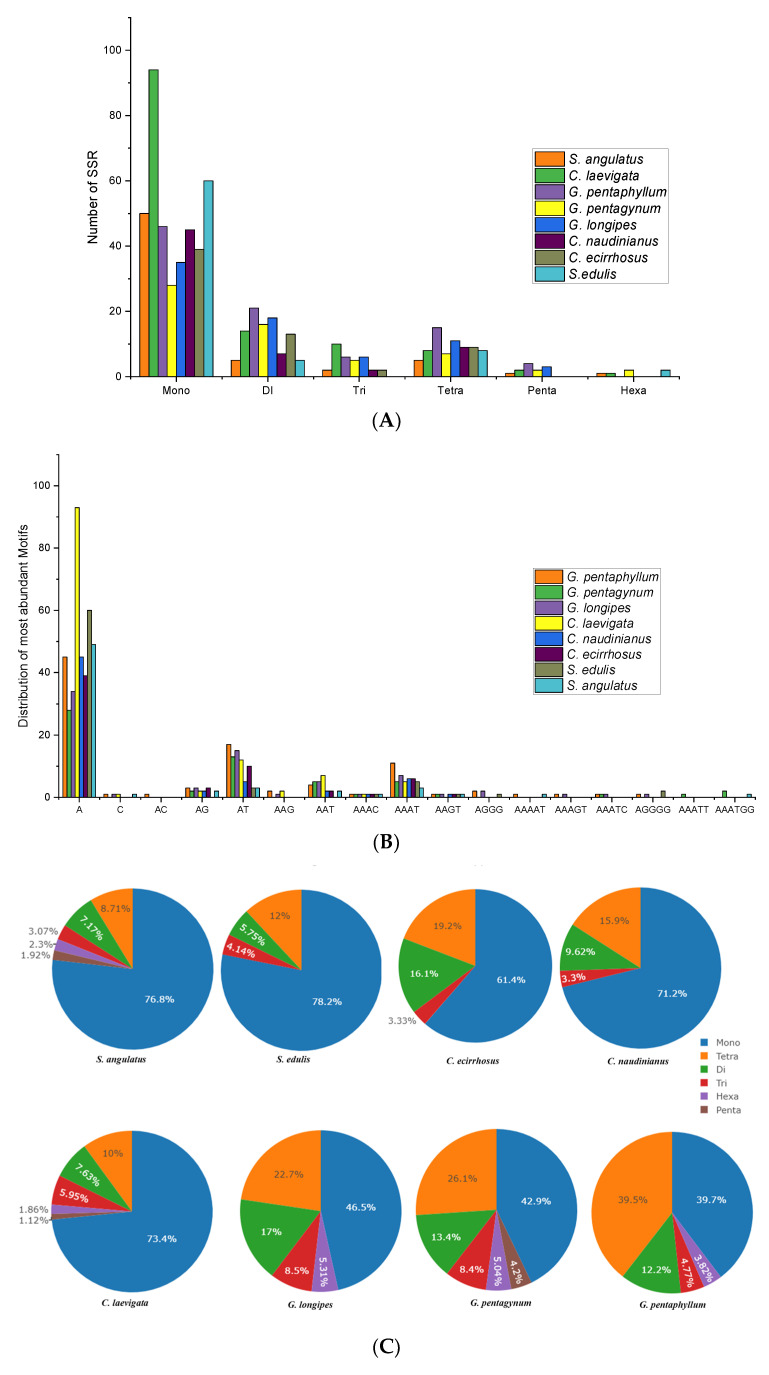
Types of SSRs (simple sequence repeats) in eight species of Cucurbitaceae family. (**A**) Distribution of various SSRs types. (**B**) The abundance of each standard motif category for each type. (**C**) SSR length percentage distribution for each type.

**Figure 5 genes-14-01776-f005:**
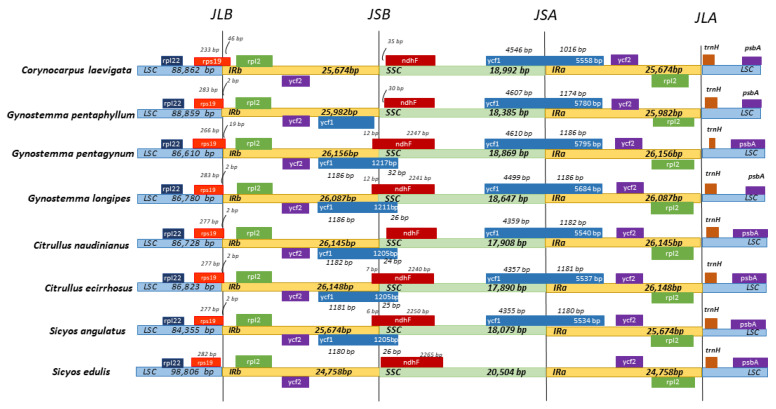
Comparison of the LSC, SSC, and IR junctions among the eight chloroplast genome sequence examined in this study.

**Figure 6 genes-14-01776-f006:**
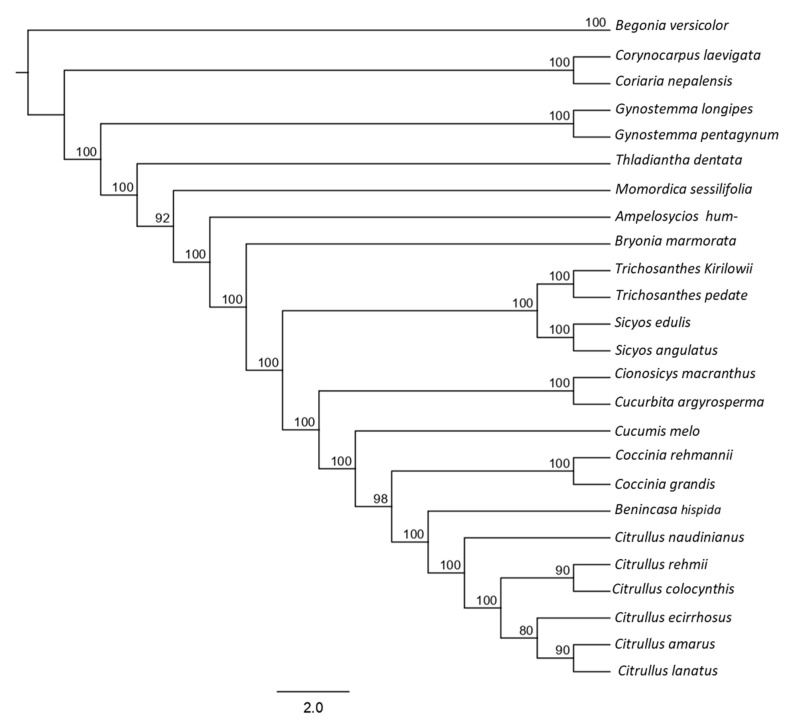
Reconstruction of phylogenetic relationship among selective representatives of the Cucurbitaceae family based on CDs sequence from complete chloroplast genomes.

**Table 1 genes-14-01776-t001:** Summary of the chloroplast genome features of eight species representing the Cucurbitaceae family.

Features	*S. angulatus*	*S. edulis*	*G.* *pentagynum*	*G. longipes*	*G.* *pentaphyllum*	*C. laevigata*	*C. ecirrhosus*	*C.* *naudinianus*
Genomic Size (bp)	154,986	154,558	157,791	157,601	159,208	159,202	157,009	156,926
LSC region (bp)	84,355	84,577	86,610	86,780	88,487	88,862	86,823	86,728
SSC region (bp)	18,079	20,504	18,869	18,647	18,385	18,992	17,890	17,908
IR region (bp)	26,276	24,758	26,156	26,087	25,982	25,674	26,148	26,145
Coding size	79,419	69,423	80,385	80,478	58,548	77,355	81,240	80,010
Total genes	129	123	133	133	132	128	133	133
No. of PCGs	84	79	87	87	79	83	89	88
tRNA (genes)	37	36	37	37	37	36	37	37
rRNA (genes)	8	8	8	8	8	8	8	8
GC content %	37.2	37.2	37.0	37.0	36.9	36.6	37.2	37.1
LSC (GC content)	35.1	35.0	34.8	34.8	34.6	34.2	35.0	34.9
SSC (GC content)	31.0	31.8	30.6	30.6	31.1	30.6	31.5	31.4
IR (GC content)	42.8	43.2	42.8	42.8	42.8	42.8	42.8	42.8
Single introns	19	19	21	19	19	20	20	19
Double introns	3	2	2	2	2	2	2	2
Duplicated genes	18	17	17	17	19	19	19	19

**Table 2 genes-14-01776-t002:** List of 74 protein coding genes shared among eight species of Cucurbitaceae family.

*accD*	*atpA*	*atpB*	*atpE*	*atpH*	*atpI*	*ccsA*
*cemA*	*clpP*	*matK*	*ndhA* ^(3)^	*ndhB* ^(1)^	*ndhC*	*ndhE*
*ndhF*	*ndhG*	*ndhH*	*ndhI*	*ndhJ*	*ndhK*	*petA*
*petB*	*petD*	*petG*	*petL*	*petN*	*psaA*	*psaC*
*psaC*	*psaI*	*psaJ*	*psbA*	*psbB*	*psbC*	*psbD*
*psbE*	*psbD*	*psbH*	*psbI*	*psbJ*	*psbK*	*psbL*
*psbM*	*psbN* ^(2)^	*psbT*	*psbZ*	*rbcL*	*rpl2* ^(1)^	*rpl14*
*rpl16*	*rpl20*	*rpl22*	*rpl23* ^(1)^	*rpl32*	*rpl33*	*rpl36*
*rpoA*	*rpoB*	*rpoc1*	*rpoc2*	*rps2*	*rpsu3*	*rpsu4*
*rps7* ^(1)^	*rps8*	*rps11*	*rps12* ^(4)^	*rps14*	*rps15*	*rps16*
*rps18*	*rps19*	*ycf3* ^(5)^	*ycf4* ^(5)^			

^(1)^ duplicated genes in all species; ^(2)^ *pbf1* in *S. angulatus* genome, but annotated as *psbN* in other species; ^(3)^ single copy in most species but duplicated in *C. ecirrhosus* genome; ^(4)^ not duplicated in *S. angulatus* genome; ^(5)^ in *S. angulatus* genome, *ycf3* is annotated as *pafl* and *ycf4* is annotated as *pafll*.

**Table 3 genes-14-01776-t003:** Tandem repeats distribution among eight species of Cucurbitaceae family.

	Intergenic Space	Coding Sequence
*S. angulatus*	22	7
*C. levigate*	60	9
*G. pentaphyllum*	74	7
*G. pentagynum*	24	8
*G. longupes*	21	9
*C. naudinaus*	25	6
*C. ecirrhosus*	13	9
*S. edulis*	16	7

## Data Availability

All other relevant data are available from the corresponding author upon reasonable request.

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
