# Peer review of "Comparative Analysis of the Chloroplast Genome of Sicyos angulatus with Other Seven Species of Cucurbitaceae Family"

_genes, 2023, doi:10.3390/genes14091776_

Round 1
Reviewer 1 Report
Kousar and Park compared chloroplast genome based on open-sourced sequences between seven different species in Cucurbitaceae family, including Sicyos Angulatus. Burcucumber is a representative and toxic invasive weed plant in many countries, which requiring physiological and genetic studies to prevent wider spreader on crop farm area. Therefore, their aims of comparative analysis between cp genome of several species in Curcurbitaceae family look an interesting research item.
However, current manuscript seems to require more considerations to publish somewhere. I hope to share my opinions with them to finish better publication.
1) Most of all, what is advances or impactful results in this study compared to Choi and et al’s chloroplast study published recently (refer to #25 reference in this manuscript)? This group finished well cp genome analysis from the same species and phylogenetic analysis with cp genomes of more numbered species in Curcurbitaceae family.
2) Why were these seven species selected for this study? Are these species sharing the same habitat or similar/different ecological features etc.? If they found interesting phenotyping data or physiological data relating to diversity of cp sequence, it must be very impactful to explain current observation. I suggest referring to Lee’s result (2022, Front.Plant Sci. 13:997521) for this concern.
3) What kinds of impact did they infer from phylogenetic tree? Is the phylogenetic difference related to physiological features or cp genome difference which they analyzed in cp genome analysis? Why don’t they discuss detailed relationships between phylogenetic analysis results with other cp genome diversity?
Minor suggestions:
1) I don’t understand Table 2. They need corrections and explanations what they want to talk in Table 2.
2) Species scientific names are always italicized through all sentences in this manuscript.
Author Response
Thank you for giving me the opportunity to submit a revised draft of my manuscript titled ‘Comparative Analysis of the Chloroplast Genome of Sicyos genus with other genera of Cucurbitaceae Family’ to ‘Genes’. I appreciate the time and effort that you and reviewers have dedicated to providing your valuable feedback on my manuscript. I am very grateful to the reviewers for their insightful comments on my paper. I have been able to incorporate the changes to reflect most of the suggestions provided by the reviewers. I have highlighted the changes within the manuscript.
Here is a point-by-point response to the reviewer’s comments and concerns.
Comments from Reviewer 1.
Comment 1: Most of all, what is advances or impactful results in this study compared to Choi and et al’s chloroplast study published recently (refer to #25 reference in this manuscript)? This group finished well cp genome analysis from the same species and phylogenetic analysis with cp genomes of more numbered species in Curcurbitaceae family.
Response: Basically, it appears that this research differs from other references studies in some respects. The reference study concentrated on sicyos angulatus characteristics and phylogenetic comparison with numerous Cucurbitaceae species. This study offers a thorough grasp of the sizes, gene contents, and structural organization of the chloroplast genomes of the sicyos and other genera belonging to the same family.
Comment 2: Why were these seven species selected for this study? Are these species sharing the same habitat or similar/different ecological features etc.? If they found interesting phenotyping data or physiological data relating to diversity of cp sequence, it must be very impactful to explain current observation. I suggest referring to Lee’s result (2022, Front.Plant Sci. 13:997521) for this concern.
Response: Eight species were chosen for this investigation after taking into account a number of variables, including their applicability to the study's goals and the accessibility of their entire chloroplast sequence in public databases. The selected species were chosen to represent various genera within the Cucurbitaceae family and span a variety of taxonomic diversity. Moreover, we choose these species because they were completely annotated on NCBI. Because these species exhibit interesting ecological traits and phenotypic diversity, despite the fact that they do not share the same habitat, makes them useful for comparative research. We want to learn more about the genetic differences and evolutionary relationships within the family by examining the chloroplast genomes of these various species. Despite the fact that our work does not primarily focus on phenotyping data, the analysis of chloroplast sequences reveals important details regarding the genetic diversity and evolutionary history of these species.
Comment 3: What kinds of impact did they infer from phylogenetic tree? Is the phylogenetic difference related to physiological features or cp genome difference which they analyzed in cp genome analysis? Why don’t they discuss detailed relationships between phylogenetic analysis results with other cp genome diversity?
Response: The inferred impact from the phylogenetic analysis is that sicyos genus shows close relationship with the Citrullus genus. The phylogenetic differences observed in the tree are likely to both physiological features and differences in the analyzed chloroplast genome. The study's limitations in terms of its scope and emphasis may account for the lack of a thorough examination of the connections between the results of the phylogenetic analysis and other cp genome diversities. Instead of in-depth discussion of the link with other cp genome diversities, the primary goal may have been to investigate the cp genomes of the chosen species and discover their phylogenetic relation within the Cucurbitaceae family.
Minor suggestions:
Comment 4: I don’t understand Table 2. They need corrections and explanations what they want to talk in Table 2.
Response: we corrected information in table 2, and briefly explained the information related to table 2 in manuscript.
Comment 5: Species scientific names are always italicized through all sentences in this manuscript.
Response: Manuscript was revised according to your comment.
Reviewer 2 Report
I am sending my comments in the attached .pdf file.

There are some areas of the text that need English editing. The current form makes it sometimes difficult to follow or sentences are clumsy. I suggest applying for a professional English editing service as in its current form it does not meet the standards of Genes journal.
Author Response
Thank you for giving me the opportunity to submit a revised draft of my manuscript titled ‘Comparative Analysis of the Chloroplast Genome of Sicyos genus with other genera of Cucurbitaceae Family’ to ‘Genes’. I appreciate the time and effort that you and reviewers have dedicated to providing your valuable feedback on my manuscript. I am very grateful to the reviewers for their insightful comments on my paper. I have been able to incorporate the changes to reflect most of the suggestions provided by the reviewers. I have highlighted the changes within the manuscript.
Here is a point-by-point response to the reviewer’s comments and concerns.
Comments from Reviewer 2.
Comment 1: Replace “Sicyos Angulatus” with “Sicyos angulatus”.
Response: Thank you for pointing this out. I agree with this comment. Therefore, we updated the manuscript as your comment.
Comment 2: L.9-10, Replace “This study aims to assemble and annotate the chloroplast genome of S. angulatus with the other species of Cucurbitaceae family” with “This study aims to assemble and annotate the chloroplast genome of S. angulatus and then compare it with plastomes of the other species representing Cucurbitaceae family.”
Response: The manuscript was revised according to your comment, L.9-11.
Comment 3: Replace “also consist of” with “and”.
Response: The manuscript was revised according to your comment, L.12
Comment 4: L. 12-13, Replace “Compared to other species, genomic size of S. angulatus is smaller almost 4,222bp smaller than G. pentaphyllum” with “Compared to other Cucurbitaceae species, chloroplast genome of S. angulatus is almost 4,222bp smaller than plastome Gynostemma pentaphyllum”. * Why did you choose Gynostemma pentaphyllum for that comparison?
Response: The manuscript was revised according to your comment, L. 13-14. In addition, we made a comparison with G. pentaphyllum because it has a greater genome size among eight species we used. Therefore, we aim to emphasize how much shorter S. angulatus is than G. pentaphyllum.
Comment 5: “All species “– what species, how many?
Response: The manuscript was revised according to your comment by mentioning number of species that we choose from Cucurbitaceae family L.15
Comment 6: L.14, Replace “have identical tRNA” with “ have the identical set of tRNA genes” * replace “C. laevigata” with “Citrullus laevigata” as it is used for the first time * replace “contain” with “contains”.
Response: The manuscript was revised according to your comment L.15.
Comment 7: L.15, Use the italic font for gene names/symbols.
Response: The manuscript was revised according to your comment, L.16.
Comment 8: L.17, “The largest tandem repeat”??? or should be rather “the highest number of tandem repeats” The reader is not sure whether you are writing here about the number of tandem repeats or the size of the repeat motif.
Response: The manuscript was revised according to your comment, by mentioning the highest number of tandem repeats, L. 18.
Comment 9: L.18-19, Replace “divergence analysis and phylogenetic tree indicate the S. angulatus is more closely resemble to genus Citrullus as compared to genus Gynostemma” with “divergence analysis and topology of the phylogenetic tree indicates that S. angulatus is more similar to genus Citrullus than to genus Gynostemma”.
Response: The manuscript was revised according to your comment L. 19-20.
Comment 10: L.19, Replace “generating” with “developing”.
Response: The manuscript was revised according to your comment L. 21.
Comment 11: L.25, Replace “Sicyos Angulatus (SA) commonly called Bur-Cucumber is an annual summer climbing 25 vines in the gourd family, Cucurbitaceae, belongs to eastern part of North America” with “Sicyos angulatus (SA) commonly called Bur Cucumber is an annual vine plant in the gourd family, Cucurbitaceae, native to the eastern part of North America”.
Response: The manuscript was revised according to your comment L. 27.
Comment 12: L.29, „different east state of…” did you mean here “various eastern parts of ...”?
Response: The manuscript was revised according to your comment L. 32.
Comment 13: L.30, Use the italic font for the Latin names of the species or genus.
Response: The manuscript was revised according to your comment L. 33.
Comment 14: L.31-32, Replace “Sicyos Angulatus” with “Sicyos angulatus”.
Response: The manuscript was revised according to your comment L. 34
Comment 15: L.32-33, Replace “As well as SA extracts show the inhibitory effect of...” with “SA extracts show also the inhibitory effect of...”.
Response: The manuscript was revised according to your comment L. 35-36
Comment 16: L.34, Explain the abbreviation “HFD-induced.”
Response: The manuscript was revised according to your comment L. 37.
Comment 17: L.35, Replace “has been noticed” with “has been reported”.
Response: The manuscript was revised according to your comment L. 38.
Comment 18: L.40, Replace “such as molecular markers, barcode identification” with “such as a source of molecular markers, DNA barcode identification”.
Response: The manuscript was revised according to your comment L. 48.
Comment 19: L.41, “due to its valuable information of a very conservative nature” * something is wrong with this sentence (clumsy) * there are far more properties that makes cp genome an interesting and valuable object of various molecular studies, add information about these features and improve that sentence.
Response: The manuscript was revised according to your comment L. 47-57.
Comment 20: L.43-46, This fragment needs improvement. My suggestion is given below: In general, the chloroplast genome size usually ranges from 120 to 160 kb, it exhibits a highly conserved circular quadripartite structure, which includes four regions: a large single copy (LSC), small single copy (SSC), and two inverted repeat regions (IR) [16,17].
Response: Agree, we changed the manuscript as your comment. L. 58-62
Comment 21: L.47 “increased in tandem”?
Response: The manuscript was revised according to your comment L. 63-64.
Comment 22: L.51 Use the italic font for the Latin names of the species or genus.
Response: The manuscript was revised according to your comment L. 67.
Comment 23: L.51-52 Replace “to solve the issues of the evolution, phylogenetic distribution, and relation among other species.” with “to solve the issues of the genus evolution and its phylogenetic relationships with other taxa”.
Response: The manuscript was revised according to your comment L. 68-70.
Comment 24: L.54-56 This fragment needs improvement. My suggestion is given below: The aims of the study are (I) to characterize the structure of the newly sequenced chloroplast genome of S. angulatus, (II) to perform the comparative analysis of the cp genomes of S. angulatus and selected representatives of Cucurbitaceae family, and (III) to reconstruct phylogenetic relationships within Cucurbitaceae family based on plastome sequences for selected species.
Response: The manuscript was revised according to your comment L. 74-82.
Comment 25: L.58-59 This sentence needs improvement. My suggestion Is given below: This data will be helpful to develop the genomic markers required for the realization of future genetic studies.
Response: The manuscript was revised according to your comment L. 82-83.
Comment 26: L.54-56 This fragment needs improvement. My suggestion is given below: The aims of the study are (I) to characterize the structure of the newly sequenced chloroplast genome of S. angulatus, (II) to perform the comparative analysis of the cp genomes of S. angulatus and selected representatives of Cucurbitaceae family, and (III) to reconstruct phylogenetic relationships within Cucurbitaceae family based on plastome sequences for selected species.
Response: The manuscript was revised according to your comment L. 74-82
Overall, we tried to extend the introduction as your comment. Moreover, we tried to explain each comment and modify the introduction section accordingly.
Comment 27: L.62-66 Please include here the information that you sequenced the cp genome of S. angulatus. As a consequence, you obtained only six genomes from GenBank.
Response: The sequences of cp genome of S. angulatus was done by DNAlink sequencing lab, South Korea (https://dnalink.com/)
Comment 28: L.67-68 Please describe more precisely the cp genome assembly and annotation procedure with the initial quality check of raw data.
Response: The cp genome assembly and annotation procedure with the initial quality check of raw data was done by DNAlink sequencing lab, South Korea (https://dnalink.com/).
Comment 29: L.69 Replace “was drawn through” with “was performed in”.
Response: The manuscript was revised according to your comment L. 94.
Comment 30: L.70 Replace “in the Shuffle-LAGAN mode with 70% cut-off identity” with “with the ShuffleLAGAN mode and 70% cut-off identity.”
Response: The manuscript was revised according to your comment L. 94-95.
Comment 31: L.72 Replace “tandem repeat finder version 4.09” with “Tandem Repeat Finder version 4.09” Add also the information about parameter settings for TRF tool.
Response: The manuscript was revised according to your comment L. 96-97.
Comment 32: L.75-77 This sentence needs improvement. I give my suggestion below: Tamura-Nei distance and neighbor-joining (NJ) clustering method was used for the construction of a phylogenetic tree. * It is not clear whether you use complete chloroplast genome sequences or CDS of shared genes only for phylogenetic analyses.
Response: The manuscript was revised according to your comment L. 103-106.
As for your comment, we added SSRs analysis, L.99-102 Reference [25].
Comment 33: L.80-88 This fragment needs improvement. My suggestion is given below: S. angulatus complete CP genome has a length of 154,986bp and displays a quadripartite circular structure. It consists of a pair of IR regions that are 26,276 bp each, separated by an LSC region and an SSC region with a length of 84,355 bp and 18,079 bp, respectively. The complete genome contains 129 total genes, including eighty-four protein-coding genes (PCGs), 37tRNA, and 8rRNA genes. The LSC region contains 83 genes, the SSC region includes 14 genes, and 17 genes are duplicated in the IRs (Figure 1). The protein-coding genes account for 51.24% (79,419 bp) of the total genome, whereas the remaining regions are composed of rRNA, tRNA, intron, and intergenic spaces.
Response: The manuscript was revised according to your comment L. 109-118.
Comment 34: L.89 Replace “in which” with “out of which”.
Response: The manuscript was revised according to your comment L. 119.
Comment 35: L.91 rps12 is a trans-spliced gene that consists of three exons that are scattered within IRs and LSC regions, but you cannot say that it contains 2 introns- – check it and improve.”
Response: The manuscript was revised according to your comment L. 121.
Comment 36: L.91-94 This fragment needs improvement. My suggestion is given below: The total GC content of cpDNA was 37.2% but we observed distinct differences between the three regions of cp genome when this feature was considered. The highest GC content was found in the IR regions (42.8%) which were followed by the LSC and SSC regions (35.1% and 31.0%, respectively).
Response: The manuscript was revised according to your comment L. 121-125.
Comment 37: L.101-104 This fragment repeats the information given above in L. 62-66.”
Response: The manuscript was revised according to your comment. We removed the double information from the manuscript.
Comment 38: L.103-104 Use the italic font for the Latin names of the species or genus.
Response: we removed that section.
Comment 39: L.105-106 This fragment refers to the results which have not been presented yet. This statement could become rather an element of the discussion.
Response: Thank you for pointing this out, as your comment we removed this information.
Comment 40: L.108-112 This fragment repeats the information given above in L. 62-66 and L.101-104.
Response: The manuscript was revised according to your comment. We removed the double information from the manuscript.
Comment 41: L.116 Suddenly you insert the statement “they have almost 37% GC content” when giving the information about genetic composition – I do not think that it is a good place for this.
Response: The manuscript was revised according to your comment. We removed this information from the manuscript.
Comment 42: L.120-123 This fragment needs improvement. My suggestion is given below: The CP genome of G. pentaphyllum is the largest among the seven studied genomes (159,208 bp) and it is 4,222 bp longer than plastome of S. angulatus (Table 1).
Response: The manuscript was revised according to your comment L. 153-156
Comment 43: L.131 “some of duplicated genes have not same sequence” – so now you should write something more about it).
Response: The manuscript was revised, and information was modified in table 1.
Comment 44: L.137-140 When writing about the differences in gene content you should be always careful as the annotations of the records available in GenBank may be not perfect. Always verify such cases by yourself as mistakes in gene annotations may happen (e.g. https://doi.org/10.1093/bib/bbac416; https://doi.org/10.1186/s12864-019-5447-1). I have verified your observations described in L. 137-140 and Table 3 and below I give my comments on them: * atpF, ycf1, and ycf2 are present in the cp genome of G. pentaphyllum but were annotated as pseudogenes – in that case this information should be used in your paper. Moreover, if there is no such information in any associated paper, you should check whether this pseudogenization is the result of a premature stop codon within these genes’ sequences or their contraction * ndhD gene is annotated in the cp genome of C. laevigata (NC_014807) as a pseudogene: it is located on negative strand position 120561.122063. If there is no such information in any associated paper, you should check whether this pseudogenization is the result of a premature stop codon within these genes’ sequences or their contraction.
Response: Thankyou for pointing out this mistake, we removed this section from our manuscript.
Comment 45: L.144“multiple”??? perhaps you wanted to write “many.”
Response: The manuscript was revised according to your comment L. 187.
Comment 46: L.148 Replace “much of the divergence” with “most of the divergence.”
Response: The manuscript was revised according to your comment L. 191.
Comment 47: L.149-150 * Replace “The IR region and protein coding genes are much less divergent as compared to” with “The IR region and protein-coding sequences are more conservative compared to...” * Generally IR is more conservative that LSC and SSC. Also, coding sequences are less divergent than non-coding regions. Here you mix these two elements.
Response: The manuscript was revised according to your comment L. 192.
Comment 48: L.152-153 Use the italic font for the Latin names of the species or genus.
Response: The manuscript was revised according to your comment L. 195-196.
Comment 49: L.160 “is the largest” – not clear, should be rather “the most frequent” The whole sentence needs improvement.
Response: The manuscript was revised according to your comment L. 204.
Comment 50: L.162 “is smaller” – not clear, perhaps” less frequent”.
Response: The manuscript was revised according to your comment L. 205.
Comment 51: L.162-163 “G. pentaphyllum ranged above 214 base pair has a longest tandem repeat.” – not clear.
Response: we removed this information from the manuscript.
Comment 52: L.164 “the largest repeats”– Are you writing here about the highest number of repeats or about their size (size of repeated motif)?
Response: The manuscript was revised according to your comment. We want to mention here the highest number of repeats, L. 207-208.
Comment 53: L.172-173 Replace “boundaries of the plastomes” with “boundaries of the analysed plastomes”.
Response: The manuscript was revised according to your comment, L.233-234.
Comment 54: L.174 use the italic font for gene names/symbols * “overlap” is not a good expression here; you can say that gene span through the LSC/IRb border and ...bp from its XXX end lies within IRb region (give the appropriate length and instead of XXX the information whether it is 5’-end or 3’-end of the sequence).
Response: The manuscript was revised according to your comment, L. 235.
Comment 55: L.176 “use the italic font for gene names/symbols.
Response: The manuscript was revised according to your comment, L. 237.
Comment 56: L.172-177 “What about ndhF on the IRb/SSC border? * Describe the situation on the two other borders, i.e., JSA and JLA. * In L. 138 you wrote that G. pentaphyllum does not contain ycf1, but you show it in Figure 4.
Response: we mentioned more details regarding each border in the manuscript, L. 239-246. Moreover about G. pentaphyllum we removed this information from our manuscript.
Comment 57: L.183-184 Replace “Tamura-nei genetic distance model and Neighbor joining building method was used” with “Tamura-Nei distance and neighbor joining clustering method was used”.
Response: The manuscript was revised according to your comment, L. 252-253.
Comment 58: L.184 It is not clear whether you used complete cp genome sequences or CDS only in phylogenetic analyses.
Response: we added the information in our manuscript, L. 253-254.
Comment 59: L.186 use the italic font for the Latin names of species or genus”.
Response: The manuscript was revised according to your comment, L. 257
Comment 60: L.196 Replace “links to” with “relationships with”.
Response: The manuscript was revised according to your comment, L. 269.
Comment 61: L.199-200 “The different genomic sizes across the Cucurbitaceae species proved to be an interesting discovery.” Really? Did you expect that these species will have the same cp genome size?
Response: we removed this information from the manuscript.
Comment 62: L.201 use the italic font for the Latin names of species or genus.
Response: The manuscript was revised according to your comment, L. 274
Comment 63: L.200-201 If you analyze all cp genomes available for the family Cucurbitaceae you will see that there are many species with bigger plastome than that reported for S. angulatus. There are also a few species which have smaller plastome... So if you discuss the position of S. angulatus within the family Cucurbitaceae you should take into account far more species.
Response: Thank you for your valuable information, but we are just pointing out genomic size among the selected species that we choose in this manuscript for comparative study from the Cucurbitaceae family.
Comment 64: L.205-209 actually all four borders have conservative characteristics with only some minor variation * not the IR existence but the dynamic of its size affect the plastome size * the IR contraction or expansion affects the genes which are placed in close vicinity of LSC/IR/SSC borders, therefore the expression “some genes” is not satisfying.
Response: The manuscript was revised according to your comment, L. 280-283.
Comment 65: L.208-209 “Moreover, among the protein coding genes the G.pentaphyllum does not contain atpF, ycf1, and ycf2, while infA is present only in genus Citrullus” * first check my comments for L. 137-140 and Table 3 * there is no connection between this sentence and the sentences above from the same paragraph
Response: we removed the information of table 3 from our manuscript.
Comment 66: L.210 “numerous” – seven chloroplast genomes (for S. angulatus and six other representatives of the family Cucurbitaceae) is definitely not numerous.
Response: The manuscript was revised according to your comment, L. 284-285.
Comment 67: L.212 uses the italic font for the Latin names of species or genus.
Response: The manuscript was revised according to your comment, L. 288.
Comment 68: L.213-215 Your results should be treated with caution as it describes the comparison of only one Sicyos species with two representatives of the genus Citrillus * L. 215 not “species” but rather “genera.
Response: The manuscript was revised according to your comment, L. 290.
Comment 69: L.216-218 The cited reference [26] described only the consequences of IR expansion and contraction for plastome size. So, if you want to keep that sentence in its current form you should add a reference supporting the role of SSC and LSC region expansion and contraction on the chloroplast genome size.
Response: we adjusted this sentence according to your comment and removed the information about role of SSC and LSC region expansion and contraction on the chloroplast genome size.
Comment 70: L.219-229 in the same paragraph you are repeating information about the LSC/IR/SSC borders, e.g. L. 219-222 and 227-229 * sentences need rephrasing * in one place you write that the IRa/LSC border in the plastome of G. pentygnum is 87 bp from trnH (L.221) but in another place, the value changes to 70 bp (227).
Response: The manuscript was revised according to your comment and removed the repeated information L. 308-310.
Comment 71: L.232-234 The sentence repeats information from L. 213-215.
Response: we removed the repeated information from the manuscript.
Comment 72: L.239 “The developed genomic markers…” Did you develop any genomic markers? You have just provided the cp genome sequence for S. angulatus which may become a source of such markers.
Response: The manuscript was revised according to your comment, L. 290
Comment 73: L.213-215 Your results should be treated with caution as it describes the comparison of only one Sicyos species with two representatives of the genus Citrillus * L. 215 not “species” but rather “genera.
Response: The manuscript was revised according to your comment, L. 320-321.
Comment 74: Figure 1 caption. The caption needs improvement. My suggestion is given below: Map of S. angulatus chloroplast genome. The genes drawn inside the circle are transcribed anticlockwise, while the outside genes are transcribed clockwise. The different colors represent genes from different functional groups.
Response: The caption was revised according to your comment, L. 128-131
Comment 75: Figure 2 caption. Replace “Ref and blue” with “Red and blue”.
Response: The caption was revised according to your comment, L. 200
Comment 76: Figure 3 What values are shown on y-axis? * On the pie chart (B) give the values for both distribution types of repeats.
Response: We changed the figure 3 and mentioned about values shown on y-axis of Figure A, as well values on pie cart of Figure B L. 213-214.
Comment 77: Figure 5 Use the italic font for the Latin names of the species or genus * delete the word “chloroplast” which stands before the species name * add in the figure caption what these values presented on the dendrogram are.
Figure 5 caption the caption needs improvement. My suggestion is given below: Reconstruction of phylogenetic relationships among selected representatives of the Cucurbitaceae family based on seven complete chloroplast genomes.
Response: we changed this figure as your comment. Moreover, the figure caption was changed according to your comment.
Comment 77: Table 1 Use the italic font for the Latin names of the species or genus.
Table 1 caption the caption needs improvement. My suggestion is given below: Summary of the chloroplast genome features of seven species representing the Cucurbitaceae family.
Response: The table in this manuscript was revised according to your comment. Moreover, we changed the caption according to your comment.
Comment 78: Table 2 why the following gene names accD, atpA, atpB, atpE, atpI, and ccsA are in the heading of the table? * Are there really only 4 duplicated genes? * I do not understand the explanation of 2) used in the superscript: you wrote “(2) pbf1 in S. angulatus genome” does it mean that psbN is replaced by pbf1 in S.angulatus chloroplast genome? Please be more precise. * I do not understand the explanation of 2) used in the superscript: you wrote “(3) ecirrhosus genome” and 3) can be found in the table next to the ndhA gene * Are you sure that rps12 is not duplicated in G.pentaphyllum and S. angulatus cp genomes? (this trans-spliced gene consists of three exons and two of them can be found in IR regions) * I do not understand the explanation of 5) used in the superscript: you wrote “(5) pafl and pafll in S. angulatus genome”, 5) can be found in the table next to the ycf3 and ycf4.
Response: we changed the table design in manuscript. We try to explain in the manuscript about table 2 information in detail L. 177-180.
Comment 79: Table 3 ycf1 gene is present in G. pentaphyllum in two copies (longer and shorter) and both of them are annotated as a pseudogene, it should be checked why the longer copy was treated as a pseudogene * ycf1 is present in G. pentagynum in two copies longer and a shorter which is annotated as a ycf1-like protein – this should be check whether this is a “normal” pseudo ycf1 like in other species but the author used another name * you write that there are two ycf1 genes in the cp genomes of C. ecirrhosus, C. nandinianus and S. angulatus while there are one complete ycf1 and the other which should be annotated as a pseudo ycf1 (shorter version) - As you can see data for ycf1 gene is not correct in this table. Please check all data presented in Table 3 and improve it.
Response: we removed this table information from the manuscript.
Overall, we changed a little information in our manuscript. First of all, we added the sicyos edulis chloroplast genome information to reveal how the sicyos genus differs from other genera in the Cucurbitaceae family. Moreover, we added the SSR analysis of eight selected species.